# Suppression of Pro-Inflammatory M1 Polarization of LPS-Stimulated RAW 264.7 Macrophage Cells by Fucoxanthin-Rich *Sargassum hemiphyllum*

**DOI:** 10.3390/md21100533

**Published:** 2023-10-12

**Authors:** Seungjin Jeong, Mi-Bo Kim, Suhyeon Baek, Joowon Lee, Hyeju Lee, Bei Cao, Yongeun Kim, Lei Cao, Sanggil Lee

**Affiliations:** 1Department of Smart Green Technology Engineering, Pukyong National University, Busan 48513, Republic of Korea; wtw3737@gmail.com (S.J.); bmh46750@gmail.com (S.B.); axxx1452@naver.com (J.L.); moveone1995@naver.com (H.L.); 2Department of Food Science and Nutrition, College of Fisheries Science, Pukyong National University, Busan 48513, Republic of Korea; mibokim1120@gmail.com; 3Warshel Institute for Computational Biology, The Chinese University of Hong Kong, Shenzhen 518172, China; bcao@cuhk.edu.cn; 4Research Division of Food Functionality, Korea Food Research Institute, Jeollabuk-do 55365, Republic of Korea; kyongeun@kfri.re.kr; 5Division of Nutritional Sciences, Cornell University, Ithaca, NY 14853, USA; 6Department of Food Science and Biotechnology, Gachon University, Seongnam 13120, Republic of Korea

**Keywords:** macrophages, macrophage polarization, *Sargassum hemiphyllum*, inflammatory

## Abstract

Macrophages play an important role in managing the onset and progression of chronic inflammatory diseases. The primary objective of this study is to explore the antioxidant potential and anti-inflammatory properties of *Sargassum hemiphyllum* ethanol extract (SHE) and its fraction. SHE and its five constituent fractions were assessed for overall antioxidant capabilities and inhibitory effects on LPS-induced inflammation by modulating macrophages polarization in both RAW 264.7 macrophages and bone-marrow-derived macrophages (BMDM). Among the organic solvent fractions of SHE, the ethyl acetate fraction displayed the highest total phenolic content and total antioxidant capacity. Notably, the n-hexane (Hex) fraction showed the most substantial suppression of LPS-induced tumor necrosis factor α secretion in BMDM among the five fractions of SHE. The SHE and Hex fraction significantly reduced the heightened expression of pro-inflammatory cytokines and inflammation-inducible enzymes induced by LPS in RAW 264.7 macrophages. In particular, the SHE and Hex fraction inhibited M1 macrophage polarization by reducing the mRNA expression of M1 macrophage markers in macrophages that were polarized toward the M1 phenotype. Furthermore, the SHE and Hex fraction attenuated the induction in nuclear factor E2-related factor 2 and its target genes, which was accompanied by an alteration in antioxidant gene expression in M1-polarized BMDM. The findings suggest that both SHE and its Hex fraction exhibit inhibitory effects on LPS-triggered inflammation and oxidative stress by modulating the polarization of M1 macrophages within macrophage populations.

## 1. Introduction

Macrophages hold a pivotal role within the body’s internal immune and inflammatory response systems that maintain homeostasis in intracellular metabolism and tissue damage and repair [1,2]. Also, macrophages are the most heterogeneous and flexible immune cells due to their response characteristics to environmental stimuli [3]. This functional conversion or adaptation of macrophages is called macrophage polarization [4]. Polarized macrophages are typically classified into two primary groups: pro-inflammatory M1 referred to as classically activated macrophages, and anti-inflammatory M2 macrophages, known as alternatively activated macrophages [5]. Classic M1 macrophages are primarily activated by molecules such as lipopolysaccharide (LPS), interferon-γ (IFN-γ), and tumor necrosis factor α (TNFα). Conversely, alternative M2 macrophages are typically induced by interleukin-4 (IL-4), IL-10, and IL-13 [6]. M1 macrophages are characterized by their ability to generate pro-inflammatory cytokines, including TNF, IL-1β, IL-6, IL-12, nitric oxide (NO), and reactive oxygen species (ROS). In contrast, M2 macrophages are distinguished by their secretion of anti-inflammatory cytokines such as IL-10, IL-4, and IL-13 and their clearance of apoptotic cells due to their elevated phagocytic capabilities [7,8].

Macrophages increase ROS and NO production through cytokines, metabolic stress, and endoplasmic reticulum stress [9]. In particular, ROS plays an essential role in maintaining and inducing M1-type macrophage polarization [10]. Also, ROS activate pro-inflammatory signaling pathways, such as nuclear factor-κB (NFκB) and activating protein-1 (AP-1), to stimulate pro-inflammatory cytokine secretion from macrophages [9]. Thus, bioactive substances with antioxidant activity can improve chronic inflammatory diseases by reducing ROS and free radicals as well as regulating the inflammatory response of macrophages.

*Sargassum hemiphyllum*, a type of brown algae, is naturally found along the coasts of Jeju Island in South Korea, as well as in regions like Taiwan, Japan, Hong Kong, and East China. It has a history of traditional use in South Korean folk medicine for addressing inflammatory diseases [11,12,13]. Research studies have provided evidence indicating that *S. hemiphyllum* extract possesses antioxidant and anti-inflammatory properties [11,14]. *S. hemiphyllum* sulfated polysaccharide extract inhibited LPS-induced pro-inflammatory cytokines through the inhibition of NFκB nuclear translocation in RAW 264.7 macrophages [11]. Furthermore, *S. hemiphyllum* hot water extract has been shown to exhibit antioxidant activity by reducing DPPH and superoxide anion radical scavenging. Also, this hot water extract also demonstrated immunostimulatory effects by enhancing immunoglobulin M secretion in human–human hybridoma HB4C5 cells and macrophage J774.1 cells [14]. *S. hemiphyllum* extract contains fucoxanthin, a xanthophyll carotenoid, which has anti-inflammatory and antioxidant effects [15,16,17,18]. However, it remains unverified whether SH ethanol extract (SHE) possesses the capability to modulate macrophage polarization, leading to anti-inflammatory responses in macrophages. Therefore, the primary aim of this research was to examine whether SHE and its various fractions possess antioxidant properties by reducing free radicals as well as the inhibitory effects on LPS-induced inflammation by modulating macrophage polarization in both RAW 264.7 macrophages and bone-marrow-derived macrophages (BMDM).

## 2. Results

### 2.1. Extraction Yield and Total Phenolic Content (TPC)

The overall extraction yield and TPC of SHE were 2% and 7.49 ± 1.12 mg gallic acid equivalents (GAE)/g dry weight, respectively (Table 1). Among the five fractions of SHE, the water (H_2_O) fraction had the highest amount of extraction yield at 48%, followed by n-hexane (Hex, 35%), butanol (BuOH, 5%), ethyl acetate (EtOA, 5%), and chloroform (CHCl_3_, 3%). The TPC of the five fractions of SHE ranged from 48.68 to 1.76 mg GAE/g dry weight. EtOA had the highest TPC at 48.68 ± 3.84 mg GAE/g dry weight, followed by CHCl_3_ (17.49 ± 0.44), Hex (6.72 ± 1.85), BuOH (3.98 ± 1.04), and H_2_O (1.76 ± 0.15) at the lowest TPC.

### 2.2. Total Antioxidant Capacity

The total antioxidant capacity of SHE and its five fractions was assessed using three different assays: ABTS, DPPH, and FRAP assay (Table 2). SHE exhibited significant ABTS, DPPH, and FRAP radical scavenging capabilities with values of 15.02 ± 1.19 mg vitamin C equivalents (VCE)/g dry weight, 22.00 ± 3.44 mg VCE/g dry weight, and 0.12 ± 0.00 mM FeSO_4_ equivalents (FSE)/g dry weight, respectively. Among the five fractions of SHE, the EtOAc fraction demonstrated the highest ABTS radical scavenging activity, with a value of 58.35 ± 5.82 mg VCE/g dry weight. This was followed by CHCl_3_ (19.75 ± 1.52), Hex (11.60 ± 0.76), BuOH (7.30 ± 2.09), and finally, H_2_O (2.65 ± 0.76). For DPPH activity in the five fractions of SHE, EtOAc exhibited the highest activity with a value of 54.12 ± 3.19 mg VCE/g dry weight. In contrast, CHCl_3_ displayed a relatively lower DPPH radical scavenging activity, with a value 39.58 ± 2.78 mg VCE/g dry weight, followed by the Hex (21.40 ± 1.39), and BuOH (13.82 ± 2.92). The DPPH radical scavenging activity of H_2_O fraction was not detected. In terms of FRAP activity, the EtOAc fraction of SHE also displayed the highest FRAP activity among the five fractions, with a value of 0.39 ± 0.02 mM FSE/g dry weight, followed by CHCl_3_ (0.15 ± 0.01), Hex (0.13 ± 0.01), BuOH (0.07 ± 0.01), and H_2_O (0.04 ± 0.00). Therefore, the EtOA fraction demonstrated the highest ABTS and DPPH scavenging activity and FRAP activity. Given the consistent trends observed between SHE and its five constituent fractions with high TPC across all three total antioxidant assays, a Pearson correlation analysis was conducted to examine the relationship between the TPC and the results of these antioxidant assays (Table 2). The analysis revealed a strong and statistically significant positive relation between TPC and the overall antioxidant capacity as measured by ABTS (*r* = 0.95, *p* < 0.01), DPPH (*r* = 0.95, *p* < 0.01). Additionally, there was a robust positive correlation between TPC and the results of the FRAP assay (*r* = 0.88, *p* < 0.01). These findings underscore the close association between the phenolic content and the overall antioxidant capacity of SHE and its fractions, as indicated by these three different antioxidant assays.

### 2.3. Effect of SHE and Its Five Fractions on Cytotoxicity and TNFα Secretion in Macrophages

The high phenolic amount and antioxidant properties of natural sources is intricately linked to their capacity for exerting anti-inflammatory effects [19]. Therefore, we investigated whether SHE and its five fractions can inhibit LPS-induced TNFα production in RAW 264.7 macrophages. First, we measured the cell cytotoxicity of SHE and five fractions and observed that there was no significant toxicity within the concentration range of 0–100 µg/mL of SHE and its five fractions (Appendix A). Based on the cell cytotoxicity data, the following experiments were carried out using concentrations less than 100 µg/mL of SHE and its fractions. In RAW 264.7 macrophages, LPS significantly increased the production of TNFα, which was significantly reduced in a dose-dependent manner by SHE and Hex and CHCl_3_ fractions (Figure 1). However, in the three fractions, EtOA, BuOH, and H_2_O fractions, no significant effect on the production of TNFα in LPS-induced BMDM was observed. In particular, even though the EtOA fraction exhibited the highest total antioxidant capacity with high phenolic content among all the fractions, it had no significant effect on LPS-induced TNFα production.

### 2.4. Effect of SHE and Hex Fraction on LPS Induced the Expression of Pro-Inflammatory Cytokines and Inflammation-Inducible Enzymes in RAW 264.7 Macrophages

Since the Hex fraction showed a higher inhibitory effect on LPS-induced TNFα production than other SHE fractions, we further proceeded to verify the inhibitory effects of SHE and Hex fraction on inflammation stimulated by LPS in RAW 264.7 macrophages. The gene expression of pro-inflammatory cytokines, such as *Il1β*, *Il6*, and *Tnf*, exhibited a significant increase in response to LPS stimulation, whereas SHE significantly reduced the expression of the *Il1β*, *Il6*, and *Tnf* genes activated by LPS in RAW 264.7 macrophages (Figure 2a). Also, LPS stimulation significantly increased the mRNA expression of NADPH oxidase 1 (*Nox1*) and cytochrome b-245 beta chain (*Cybb*), the gene which encodes NOX2 enzyme. Both *Nox1* and *Cybb* were significantly reduced by SHE in RAW 264.7 macrophages (Figure 2b). SHE also significantly suppressed the LPS-induced mRNA expression of inflammation-inducible enzymes, such as nitric oxide synthase 2 (*Nos2*, the gene name for inducible NOS (iNOS)), and cyclooxygenase 2 (*Cox2*) in RAW 264.7 macrophages (Figure 2c). Furthermore, SHE demonstrated a noticeable reduction in the protein levels of iNOS and COX2 in RAW 264.7 macrophages activated by LPS (Figure 2d). In the case of the Hex fraction, it exhibited a similar anti-inflammatory effect to that of SHE in RAW 264.7 macrophages. LPS significantly increased the gene expression levels of the *Il1β*, *Il6*, and *Tnf* genes, which were significantly repressed by the Hex fraction in RAW 264.7 macrophages (Figure 3a). Also, upon stimulation with LPS, the gene expression of *Nox1* and *Cybb* was significantly increased. However, the Hex fraction completely eliminated the expression of the *Nox1* and *Cybb* genes in RAW 264.7 macrophages (Figure 3b). Furthermore, the Hex fraction significantly diminished both *Nos2* and *Cox2* gene expression in RAW 264.7 macrophages stimulated by LPS (Figure 3c). The elevated protein expression levels of iNOS and COX2 by LPS stimulation were markedly reduced by the Hex fraction in RAW 264.7 macrophages (Figure 3d).

### 2.5. Inhibitory Effect of SHE and Hex Fraction on M1 Macrophages Polarization in RAW 264.7 Macrophages and BMDM

Since the LPS-stimulated inflammatory response is closely linked to M1 polarization [20], we confirmed whether the anti-inflammatory effect of SHE and the Hex fraction in macrophages is due to its ability to inhibit macrophage polarization toward M1. RAW 264.7 macrophages and BMDM were induced to polarize into M1 macrophages using a combination of LPS and INF-γ. During the polarization of RAW 264.7 macrophages into M1 macrophages, the mRNA expression of M1 macrophages makers, including *Il1β*, *Il6*, *Tnf*, *Nos2*, the cluster of differentiation (*Cd86*), and chemokine CC ligand-2 (*Ccl2*, the gene name for monocyte chemoattractant protein-1), was significantly suppressed by SHE (Figure 4a). Also, BMDM macrophages’ polarization into M1 showed increased mRNA expression of *Il1β*, *Il6*, *Tnf*, *Nos2*, *Cd86*, and *Ccl2*, while SHE significantly reduced their expressions except for *Tnf* at 50 μg/mL. In the case of the Hex fraction, the LPS and INF-γ-induced expression of *Il1β*, *Il6*, *Tnf*, *Nos2*, *Cd86*, and *Ccl2* was completely attenuated by the Hex fraction in M1-polarized RAW 264.7 macrophages (Figure 5a). Furthermore, their induced expressions were also significantly alleviated by the Hex fraction in M1-polarized BMDM (Figure 5b).

### 2.6. Effect of SHE and Hex Fraction on Antioxidant Gene Expression in M1-Polarized BMDM

As we observed the reduction in M1 polarization in both RAW 264.7 macrophages and BMDM by SHE and its Hex fraction, further investigation was conducted to determine whether SHE and its Hex fraction can alter the expression of antioxidant genes in M1-polarized BMDM. BMDM were polarized into M1 macrophages by a combination of LPS and INF-γ in the absence or presence of SHE and its Hex fraction. The combination of the LPS and INF-γ-induced expression of antioxidant genes, such as glutathione peroxidase 1 (*Gpx1*), superoxide dismutase (*Sod1*), and catalase (*Cat*). Although both SHE and its Hex fraction decreased the expression of *Gpx1* and increased the expression of *Cat*, a reduced *Sod1* mRNA expression was observed in Hex-fraction-treated samples but not in SHE-treated samples (Figure 6a,b). Nuclear factor E2-related factor 2 (NRF2) is recognized for its role in safeguarding cells against oxidative stress and inflammation by increasing antioxidant enzymes that can mediate the anti-inflammatory polarization of macrophages. Therefore, we measured the *Nfe2l2* (the gene name for NRF2) and heme oxygenase-1 (*Hmox1*), an NRF2 target gene in M1-polarized BMDM. The mRNA expression of *Nfe2l2* was significantly increased by the combination of LPS and INF-γ stimulation, which was significantly restored by both SHE and its Hex fraction. The mRNA expression of *Hmox1* was significantly increased by a combination of LPS and INF-γ stimulation; while the Hex fraction inhibited the induction of *Hmox1* expression, SHE did not alter its expression.

### 2.7. Fucoxanthin Content in SHE and Hex Fraction

Since *S. hemiphyllum* is known to contain fucoxanthin, which has anti-inflammatory and antioxidant effects [15,16,17,18], we further investigated the fucoxanthin amount in SHE and its Hex fraction using high-performance liquid chromatography (HPLC) for qualitative analysis and liquid chromatography tandem mass spectrometry (LC-MS/MS) for quantitative analysis. Fucoxanthin in the SHE and the Hex fractions was detected at a retention time of 11.8 min, equivalent to the standard fucoxanthin retention time (Figure 7a). The fucoxanthin content of SHE and its Hex fraction was 1.50 ± 0.08 and 2.56 ± 0.12 mg/g dry weight, respectively, indicating that the fucoxanthin content in the Hex fraction was 1.7 times higher than that in SHE (Figure 7b).

## 3. Discussion

Macrophages are known to serve as important targets for therapeutic strategies to prevent or treat inflammation-related chronic inflammation and oxidative stress [21]. It is necessary to discover bioactive materials that improve chronic inflammatory diseases by reducing ROS and free radicals and regulating the inflammatory response of macrophages. Studies have provided evidence that *Sargassum hemiphyllum* possesses antioxidant properties along with radical scavenging activity and suppress inflammatory cytokines secretion in macrophages [11,14]. However, up until this study, the effect of SHE and its fractions on the regulation of macrophages’ polarization to induce an anti-inflammatory response in macrophages had not been firmly established. In the course of this study, we unveiled that SHE and its Hex fraction not only suppressed inflammatory cytokine gene expressions but also regulated antioxidant genes. This is at least in part due to the inhibition of M1 macrophage polarization in activated macrophages such as RAW 264.7 cells and BMDM.

Macrophages are classified into two different phenotypes based on their function or the types of stimuli they receive [6,22]. Immune cells are functionally classified into two main categories pro-inflammatory (M1) and anti-inflammatory (M2). Classic M1 macrophages are primarily activated by Th1 cytokines such as IFN-γ and TNFα. For instance, RAW 264.7 macrophages can be polarized into M1 in response to a combination of LPS and IFN-γ [23,24]. In responses to these stimuli, M1 macrophages initiate a pro-inflammatory response by releasing cytokines such as IL-6, IL-1β, and TNFα. This polarization is a crucial part of the immune response and plays a pivotal role in host defense. On the other hand, alternative M2 macrophages are activated by Th2 cytokines including IL-4, IL-10, and IL-13. When these cytokines present, M2 macrophages undergo a shift in their phenotype and begin to produce anti-inflammatory cytokines such as IL-10 and TGFβ. This transition is associated with an anti-inflammatory and tissue repair role in the immune response [6,22]. Furthermore, M1 macrophages play a pivotal role in stimulating cytotoxic adaptive immunity. To carry out their immune functions, M1 macrophages upregulate the expression of the major histocompatibility complex class II (MHC II) and co-stimulatory molecules, including CD40, CD80, and CD86 [25].

In this study, we found that the polarization of both the RAW 264.7 macrophages and BMDM macrophages into M1 macrophages stimulated the expression of *Il1β*, *Il6*, *Tnf*, *Nos2*, *Cd86*, and *Ccl2*. However, SHE and its Hex fraction inhibited M1 macrophages’ polarization by attenuating the induction of *Il1β*, *Il6*, *Nos2*, *Cd86*, and *Ccl2* mRNA expression. Noticeably, *Tnf* mRNA level in BMDM was decreased by 25 μg/mL SHE but not 50 μg/mL SHE. This may be due to its unique constitutive decay element (CDE) [26]. At high concentrations, certain substances in SHE may inhibit the degradation of *Tnf* mRNA in BMDM though CDE. However, further investigation is needed.

These results suggest that SHE and the Hex fraction have the capacity to hinder the process of macrophage M1 polarization, which is associated with pro-inflammatory functions. The inhibition of M1 polarization had a notable effect on reducing the inflammatory response provoked by the combination of LPS/IFN-γ in both RAW 264.7 and BMDM macrophages, which led to the reduced production of pro-inflammatory cytokines when these cells were exposed to the LPS and IFN-γ combination, compared to the control. Therefore, our data provide evidence that the modulatory effect of SHE and its Hex fraction on macrophage phenotype in both RAW 264.7 macrophages and BMDM contributed to the anti-inflammatory effect of SHE and its Hex fraction. Also, SHE and its Hex fraction inhibit M1 polarization in both RAW 264.7 macrophages and BMDM, which diminish the inflammatory responses that can be triggered by various stimuli commonly found in chronic inflammation disease. Hence, SHE and its Hex fraction might have therapeutic potential for alleviating chronic-inflammation-associated conditions by modulating macrophage polarization.

The strong antioxidant property of fucoxanthin is attributed to its distinct structure, characterized by the presence of oxygen functional groups like carbonyl, carboxyl, epoxy, hydroxyl, and an allene bond within its polyene hydrocarbon chain [18]. Furthermore, the existence of an allenic bond in fucoxanthin contributes to its strong capacity to effectively quench oxygen and scavenge free radicals. This allenic bond enhances its antioxidant properties and makes it effective in neutralizing ROS and free radicals [27]. In the current study, SHE and its Hex fraction contained fucoxanthin as 1.50 and 2.56 mg/g extract, respectively, which contribute to the ABTS and DPPH radicals’ scavenging activities. Also, SHE and its Hex fraction were found to significantly attenuate the overexpression of antioxidant enzyme *Gpx1* in M1-polarized macrophages. These results suggest that SHE and the Hex fraction have the potential to regulate the expression of antioxidant enzymes, counteracting the oxidative stress associated with M1 macrophage polarization. In line with our results, the potent antioxidant capacity of SHE has been substantiated in previous studies. The hot-water extract of *S. hemiphyllum* showed DPPH free radicals’ scavenging activity, superoxide anion scavenging activity, and Fe^3+^ reducing activity as IC_50_ = 1.58, 2.41, and 0.41 mg/mL, respectively, which could be attributed to high levels of total phenolic compounds [14]. The levels of SOD, CAT, and malondialdehyde (MDA) in the brain, kidney, and liver were significantly attenuated when 30 mg/kg of crude phlorotannins extracts of *S. hemiphyllum* was administrated to Kunming mice pre-treated with carbon tetrachloride (CCl_4_) [28].

It is well-established that oxidative stress resulting from an excessive production of ROS can trigger an inflammatory response [29]. LPS, a major component of the outer membrane of Gram-negative bacteria, indeed possesses the ability to induce the production of ROS when it interacts with macrophages. Simultaneously, LPS triggers the secretion of pro-inflammatory cytokines and inflammation-inducible enzymes in these macrophages [30]. In the current study, SHE and its Hex fraction were found to significantly decrease the gene expression of pro-inflammatory cytokines, including *Il1β*, *Il6*, and *Tnf* in LPS-induced macrophages. During inflammatory processes, the elevated expression of pro-inflammatory cytokines is often mediated by the action of enzymes, such as iNOS and COX2. These enzymes play a crucial role in generating NO and prostaglandins, respectively, which are signaling molecules that contribute to the inflammatory response [31]. The production of NO through peroxynitrite is recognized to amplify oxidative stress and nitration within macrophages in response to ROS [32]. We found that SHE and its Hex fraction significantly decreased LPS-stimulated *Nos2* and *Cox2* gene expression levels with a concomitant decrease in their protein expression levels in macrophages. This result suggests that the reduction in the expression of *Nos2* and *Cox2* in LPS-stimulated macrophages by SHE and its Hex fraction plays a significant role in the reduced production of pro-inflammatory cytokines. This indicates that the anti-inflammatory effect of SHE and the Hex fraction is, at least in part, mediated by their ability to downregulate the expression of these key enzymes involved in the inflammatory response, thereby limiting the production of pro-inflammatory cytokines. Furthermore, SHE and its Hex fraction demonstrated antioxidant properties by reducing the scavenging activity of DPPH and ABTS free radicals, as well as by exhibiting FRAP activity. Therefore, the total antioxidant capacity of SHE and its Hex fraction are likely to contribute to the decreased gene expression of *Il1β*, *Il6*, *Tnf*, *Nos2*, and *Cox2* in LPS-induced macrophages. Also, NOX enzymes, including NOX1, NOX2, NOX4, and NOX5, are recognized for their ability to generate ROS within macrophages. This ROS production can be a contributing factor to inflammation and is involved in various cellular processes and signaling pathways associated with immune responses and inflammation [33]. We found that SHE and its Hex fraction completely abolished the LPS-stimulated gene expression of *Nox1* and *Cybb* in macrophages. SHE and its Hex fraction significantly suppressed the mRNA expression of *Cybb*, even surpassing the level observed in cells not exposed to LPS. The NOX2 enzyme, which is encoded by the *Cybb* gene, is involved in the M1 polarization signaling and its inhibition has been proposed to be responsible for the suppression of stress-induced oxidative burst. Substances with robust antioxidant capacity have demonstrated such potent inhibitory effects in several studies [34,35]. Although this study did not directly measure the effect of SHE and its Hex fraction on ROS generation in macrophages, it was observed that they can reduce the increased production of ROS by inhibiting *Nox1* and *Cybb* expression in LPS-activated macrophages.

NRF2, indeed, plays a crucial role in the endogenous antioxidant defense system by promoting the expression of numerous antioxidant and detoxification genes [18,36,37]. Under oxidative stress conditions, ROS can trigger the dissociation of NRF2 from its inhibitor, Kelch-like ECH-associated protein 1. This dissociation allows NRF2 to translocate into the cell nucleus. Once in the nucleus, NRF2 can bind to antioxidant response elements in the DNA, promoting the transcription of antioxidant and detoxification genes such as *Hmox1* and *Sod1* [38]. We found in the current study that M1 polarization significantly increased the mRNA expression of *Nfe2l2*, which was significantly suppressed by SHE and its Hex fraction. Hex fraction, but not SHE, reduced the expression of *Hmox1* and *Sod1*, two target genes regulated by NRF2. It is plausible that other substances in SHE may interfere with this signaling pathway. Additionally, the activation of NRF2 and the subsequent elevation of antioxidant gene expression can, indeed, have an anti-inflammatory effect by preventing the activation of NFκB [38]. Lee et al. [39] demonstrated that the berry anthocyanin fraction had significant anti-inflammatory and antioxidant effects. It reduced the mRNA expression of pro-inflammatory cytokines and lowered cellular ROS production in LPS-induced BMDM from wild-type mice. In contrast, when the same berry anthocyanin fraction was tested on LPS-stimulated BMDM from mice lacking *Nfe2l2*, it still significantly decreased the mRNA expression of pro-inflammatory cytokines without reducing cellular ROS levels. Furthermore, it has been demonstrated that fucoxanthin, which is abundant in *S. hemiphyllum*, inhibits LPS-induced inflammation and oxidative stress in macrophages by regulating nuclear factor NRF2 through the phosphatidylinositol 3-kinase/AKT pathway [18]. In the context of this study, SHE and its Hex fraction significantly reduced LPS-stimulated TNF-α secretion and the expression of *Il1β*, *Il6*, and *Tnf*, which is likely attributed to NRF2 regulation in M1-polarized BMDM. Therefore, the presence of fucoxanthin in SHE and its Hex fraction may contribute to suppressing the inflammatory response effect through the nuclear translocation of NRF2 in macrophages. Other components, such as phenolic compounds and meroterpenoids, are likely to contribute to their anti-inflammatory effect as well. A comprehensive understanding of the molecular mechanism of the NRF2-mediated antioxidant, and the anti-inflammatory effects of SHE in polarized macrophages, is warranted in further study.

## 4. Materials and Methods

### 4.1. Preparation of the Extraction and Fractions

*Sargarssum hemiphyllum* collected from Yere-dong, Seogwipo-si, Jeju Island, Republic of Korea was purchased from Parajeju (Jeju Island, Republic Korea). SH was washed two or three times with tap water, blended, and stored in a plastic bag at −20 °C for future use. The blended *S. hemiphyllum* was extracted twice with 100% ethanol by sonication for 1 h at room temperature. *S. hemiphyllum* extract was filtered using filter paper (F1093 grand, Chmlab, Barcelona, Spain). Then, the *S. hemiphyllum* extract was concentrated using a rotary vacuum evaporator (Büchi Labortechnik AG, Flawil, Switzerland) at a temperature 40 °C. Following concentration, the extract was dissolved in deionized water and underwent liquid–liquid partitioning using solvents of varying polarity. The aqueous crude extract was subjected to this partitioning process two or three times with an equal volume of Hex, CHCl_3_, EtOAc, BuOH, and H_2_O. Subsequently, the resulting fractions were, once again, concentrated, utilizing the rotary vacuum evaporator (Büchi Labortechnik AG) to quantify the yield obtained from each fraction.

### 4.2. Total Phenolic Contents and Total Antioxidant Capacity

TPC and total antioxidant activities, including ABTS, DPPH, and FRAP assay of SHE and its five fractions, were evaluated as previously described in our methods [40,41,42]. TPC was quantified and expressed as mg GAE/g dry weight for both SHE and its respective fractions through comparison with a calibration curve established using standard gallic acid. For DPPH and ABTS radical scavenging activities, the results were expressed as mg VCE/g dry weight both for SHE and for each fraction of SHE. FRAP values were expressed as mM FSE/g dry weight for both SHE and its individual fractions.

### 4.3. Reagents and Cell Culture

Dulbecco’s modified Eagle’s medium (DMEM, high glucose), thiazolyl blue tetrazolium bromide (MTT), and LPS (*Escherichia coli* O26:B6) were purchased from Sigma-Aldrich (St. Louis, MO, USA). Fetal bovine serum (FBS) and penicillin–streptomycin solution was purchased from WelGENE (Gyeongsan, Republic of Korea) and Hyclone (Logan, UT, USA), respectively. Interferon gamma (INF-γ) was purchased from NKMAX (Sungnam, Republic Korea).

RAW 264.7 macrophages were obtained from the American Type Culture Collection (Manassas, VA, USA). These macrophages were cultured in high-glucose Dulbecco’s modified Eagle’s medium (DMEM), which was supplemented with 10% fetal bovine serum (FBS) and antibiotics consisting of 100 units/mL of penicillin and 100 µg/mL of streptomycin. The culture conditions maintained these cells at a temperature of 37 °C in a humidified atmosphere containing 5% CO_2_. Mouse BMDM were isolated from the femoral and tibial bone marrow of 15-week-old male C57BL/6N mice. Briefly, the isolated bone marrow cells were differentiated into macrophages in BMDM medium consisting of macrophage colony-stimulating factor, 5 mM L-glutamine, 10% FBS, penicillin (100 U/mL), and streptomycin (100 µg/mL).

### 4.4. Cytokine Measurement

SHE and five fraction samples were dissolved in dimethyl sulfoxide (DMSO) and diluted 1:1000 in medium. BMDM were pretreated with SHE and its five organic solvent fractions at 0, 25, 50, and 100 µg/mL for 6 h and then stimulated with 100 ng/mL of LPS for 24 h. The culture media were collected after BMDM were treated with SHE and five organic fractions. TNFα levels were measured using an enzyme-linked immunosorbent assay using a TNFα mouse uncoated ELISA kit (Invitrogen, Carlsbad, CA, USA) according to the manufacturer’s instructions.

### 4.5. Cell Viability

Cell viability was measured using colorimetric MTT assay. RAW 264.7 macrophages were seeded into a 96-well plate at a density of 1 × 10^4^ cells/well and treated with various concentrations (0–400 µg/mL) of SHE and its five organic solvent fraction. After sample treatment for 24 h, MTT solution at a concentration of 500 μg/mL was introduced to the cells and allowed to incubate at a temperature of 37 °C for a duration of 90 min. Following the incubation with MTT solution, the MTT solution was removed from the cells. The insoluble formazan dye, which had formed as a result of cellular metabolic activity, was then dissolved by using dimethyl sulfoxide. Subsequently, the absorbance of the solution was measured at a wavelength of 560 nm spectrophotometrically, utilizing a spectrophotometer from Thermo Fisher Scientific (Waltham, MA, USA). This measurement allows for the assessment of cell viability and metabolic activity based on the conversion of MTT into formazan by viable cells.

### 4.6. Quantitative Real-Time PCR (qRT-PCR)

RAW 264.7 macrophages for pro-inflammatory cytokines and antioxidant genes analysis were treated with SHE and Hex fraction at 0, 25, and 50 µg/mL for 6 h and then induced by 100 ng/mL of LPS for 3 h in the presence or absence of SHE and its Hex fraction. For M1 polarization experiments, RAW 264.7 macrophages and BMDM were treated with SHE and its Hex fraction at 0, 25, and 50 µg/mL for 6 h and then induced by 100 ng/mL of LPS and 50 ng/mL of INF-γ for 24 h in the presence or absence of SHE and its Hex fraction. Total RNA was extracted from macrophages using homemade Trizol reagent, and cDNA synthesis and qRT-PCR analysis using the SYBR Green Q-PCR Master Mix (Smart Gene, Daejeon, Republic of Korea) and QuantStudio™ 1 Real-Time PCR system (Thermo Fisher Scientific) were conducted as previously described in our methods [41,42]. The primers used in this study are listed in Appendix A.

### 4.7. Western Blot Analysis

RAW 264.7 macrophages for Western blot analysis were pretreated with SHE and its Hex fraction at 0, 25, and 50 µg/mL for 6 h and then induced by 100 ng/mL of LPS for 24 h in the presence or absence of SHE and its Hex fraction. Macrophages were lysed using CETi lysis buffer (TransLab, Daejeon, Republic Korea). The lysate protein concentrations were measured by using the Pierce™ BCA protein assay kit (Thermo Fisher Scientific). Western blot analysis was performed as previously described in our methods [41,42].

### 4.8. High-Performance Liquid Chromatography Analysis

The dried SHE and its Hex fraction samples were dissolved in methanol then the content of fucoxanthin was analyzed using an Agilent Technologies 1200 Series HPLC system (Agilent, Palo Alto, CA, USA) with a binary pump, diode array detector (G1315B), and C18 reverse-phase symmetry analytical column (5 μm × 250 mm × 4.6 mm, YMC, Koyoto, Japan). The temperature of the column was 35 °C, and the mobile phase was HPLC grade acetonitrile with 0.1% formic acid (solvent A) and distilled water with 0.1% formic acid (solvent B). The gradient of the mobile phase used was as follows: (solvent A: solvent B) 0 min (80:20), 2 min (80:20), 12.5 min (100:0), 15 min (100:0), 15.1 min (80:20), and 18 min (80:20). The flow rate was 1 mL/min, and the injection volume was 5 µL. The UV-Vis spectra were detected at 447 nm for fucoxanthin qualitative.

The total fucoxanthin content of SHE and its Hex fractions for quantitative study was analyzed using a Xevo TQ-MS triple quadrupole mass spectrometer (Waters, Guyancourt, France) equipped with a Waters Acquity UPLC system (Waters). Chromatographic separation was achieved on a Waters BEH C18 column (1.7 μm × 2.1 mm × 100 mm, Waters). The analytical conditions were as follows: electrospray ionization with positive mode, desolvation temperature 500 °C, desolvation gas flow rate 700 L/h, and source temperature 150 °C. Acetonitrile containing 0.1% formic acid (solvent A) and distilled water containing 0.1% formic acid (solvent B) were used for the mobile phases. The gradient condition was as follows: (solvent A: solvent B) 0–1 min (20:80), 1–10 min (95:5), 10–15 min (95:5), 15–16 min (20:80), and 16–20 min (20:80). The column was maintained for 2.5 min with 95% (B). Mass spectrometric analysis was performed in the positive ion mode with a cone voltage of 35 V, collision energy of 30 V, and a dwell time of 0.091 s. The MRM transitions of fucoxanthin (*m*/*z*) were 659 > 109 (Table 3). The total fucoxanthin content in SHE and its Hex fraction was measured by using a standard curve and expressed as mg/g dry weight.

### 4.9. Statistical Analysis

All analyses were repeated three times. One-way analysis of variance (ANOVA) and Tukey’s post hoc test were performed using GraphPad 9.0 (GraphPad Software, La Jolla, CA, USA). All data were considered significant at *p* < 0.05. All values were expressed as mean ± SD.

## 5. Conclusions

In conclusion, this study provides evidence that both SHE and its Hex fraction possess significant inhibitory effects on LPS-activated inflammation and oxidative stress by modulating M1 macrophage polarization in macrophages. SHE and its Hex fraction suppressed the LPS-stimulated gene expression of pro-inflammatory cytokines and inflammation-inducible enzymes in macrophages, which is accompanied by TPC and overall antioxidant capacity by SHE and its Hex fraction. In particular, SHE and its Hex fraction inhibited M1 macrophages’ polarization in macrophages, leading to a reduced pro-inflammatory response. Also, SHE and its Hex fraction were observed to mitigate the LPS-stimulated induction in *Nfe2l2* and its target genes, thus contributing to the suppressive effects of SHE and its Hex fraction on oxidative stress and inflammation. Therefore, based on our findings, it is suggested that SHE holds significant promise as a nutraceutical material with the potential for preventing and treating chronic inflammatory diseases. This potential is attributed to its ability to modulate M1 macrophage polarization, a pivotal factor in the development of these diseases.

## Figures and Tables

**Figure 1 marinedrugs-21-00533-f001:**
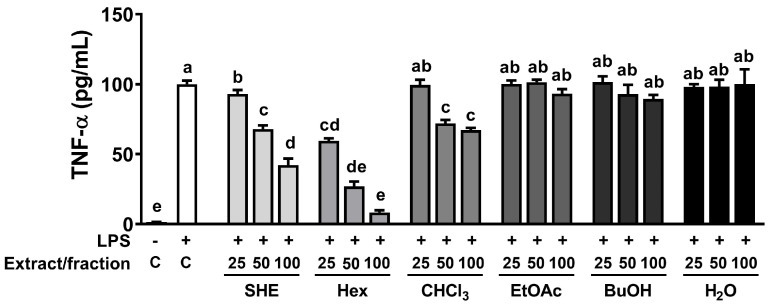
The effects of SHE and its five fractions on TNF-α secretion in LPS-induced RAW 264.7 macrophages. Cells were treated with SHE and its five fractions at concentrations of 0, 25, 50, 100 µg/mL for 6 h and then stimulated by LPS (100 ng/mL) for 24 h in the presence or absence of SHE and its five fractions to measure TNF-α secretion. Columns that do not share a common letter are significantly different (*p* < 0.05). The data are presented as means ± SD.

**Figure 2 marinedrugs-21-00533-f002:**
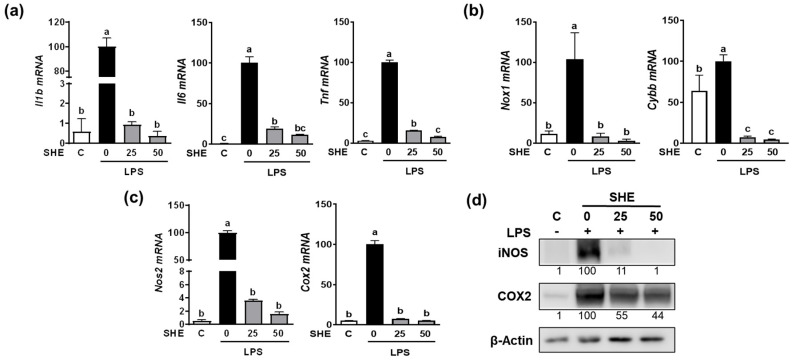
The effect of SHE on LPS-induced pro-inflammatory cytokines and inflammation-inducible enzymes in RAW 264.7 macrophages. (**a**–**c**) RAW 264.7 macrophages were first pre-exposed to SHE at different concentrations (0, 25, and 50 µg/mL) for 6 h and then stimulated by LPS at a concentration of 100 ng/mL for 3 h in the presence or absence of SHE for gene analysis by RT-PCR. *Rpl32* was utilized as an internal control. (**d**) RAW 264.7 macrophages were first pre-exposed to SHE at different concentrations (0, 25, and 50 µg/mL) for 6 h and then stimulated by LPS at a concentration of 100 ng/mL for 24 h in the presence or absence of SHE for Western blot analysis. β-Actin was utilized as a loading control. Columns that do not share a common letter are significantly different (*p* < 0.05). The data are presented as means ± SD.

**Figure 3 marinedrugs-21-00533-f003:**
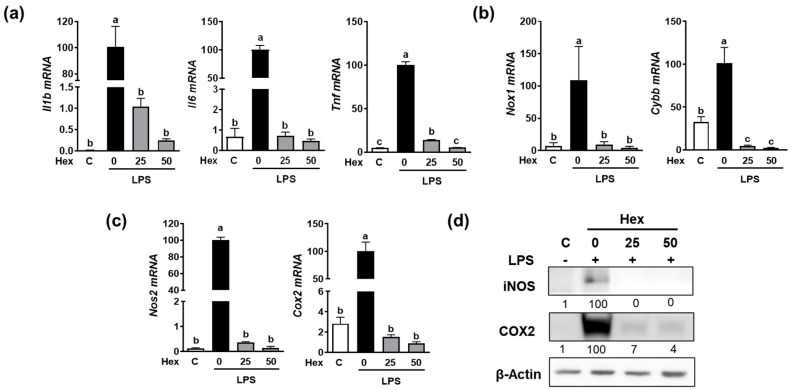
The effect of Hex fraction of SHE on LPS-induced pro-inflammatory cytokines and inflammation-inducible enzymes in RAW 264.7 macrophages. (**a**–**c**) RAW 264.7 macrophages were first pre-exposed to Hex fraction at different concentrations (0, 25, and 50 µg/mL) for 6 h and then stimulated by LPS at a concentration of 100 ng/mL for 3 h in the presence or absence of Hex fraction for gene analysis by RT-PCR. *Rpl32* was utilized as an internal control. (**d**) RAW 264.7 macrophages were first pre-exposed to Hex fraction at different concentrations (0, 25, and 50 µg/mL) for 6 h and then stimulated by LPS at a concentration of 100 ng/mL for 24 h in the presence or absence of Hex fraction for Western blot analysis. β-Actin was utilized as a loading control. Columns that do not share a common letter are significantly different (*p* < 0.05). The data are presented as means ± SD.

**Figure 4 marinedrugs-21-00533-f004:**
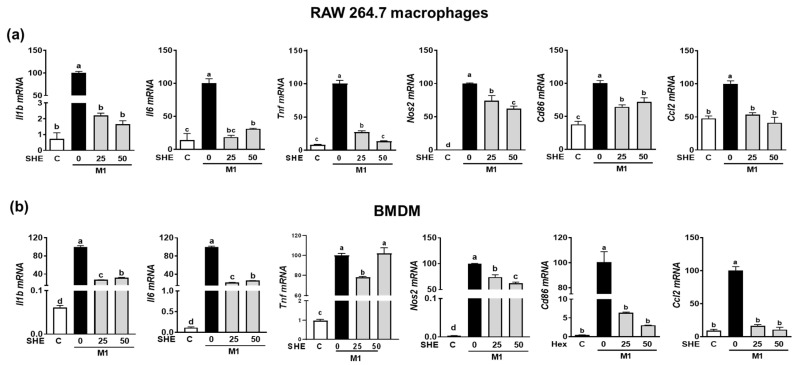
The effect of SHE on M1 macrophages polarization in RAW 264.7 macrophages and BMDM. (**a**) RAW 264.7 macrophages and (**b**) BMDM were subjected to a pre-exposed to SHE at different concentrations (0, 25, and 50 µg/mL) for 6 h. Subsequently, the cells were stimulated with a combination of LPS (100 ng/mL) and IFN-γ (50 ng/mL) for 24 h with or without SHE for gene analysis by RT-PCR. *Rpl32* was utilized as an internal control. Statistical significance was indicated by different letters within the same column (*p* < 0.05). The data are presented as means ± SD.

**Figure 5 marinedrugs-21-00533-f005:**
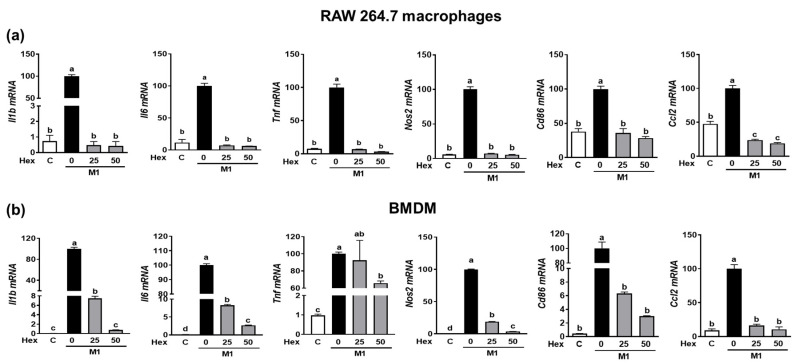
The effect of Hex fraction of SHE on M1 polarization macrophages in RAW 264.7 macrophages and BMDM. (**a**) RAW 264.7 macrophages and (**b**) BMDM were subjected to a pre-exposed Hex fraction at concentrations of 0, 25, and 50 µg/mL for 6 h. Subsequently, the cells were stimulated with a combination of LPS (100 ng/mL) and IFN-γ (50 ng/mL) for 24 h, with or without Hex fraction, for gene analysis by RT-PCR. *Rpl32* was employed as an internal control. Statistical significance was indicated by different letters within the same column (*p* < 0.05). The data are presented as means ± SD.

**Figure 6 marinedrugs-21-00533-f006:**
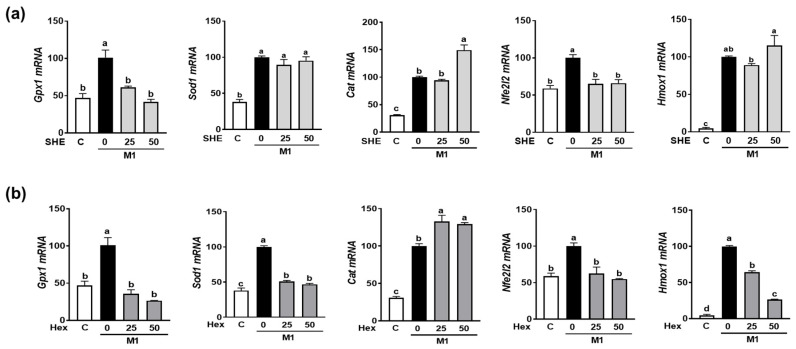
The effect of SHE and its Hex fraction on antioxidant gene expression in M1-polarized BMDM. BMDM were treated with (**a**) SHE or (**b**) Hex fraction at 0, 25, and 50 µg/mL for 6 h and then stimulated by LPS (100 ng/mL) and IFN-γ (50 ng/mL) for 24 h with or without SHE and Hex fraction for gene analysis by RT-PCR. *Rpl32* was used as internal controls. Statistical significance was indicated by different letters within the same column (*p* < 0.05). The data are presented as means ± SD.

**Figure 7 marinedrugs-21-00533-f007:**
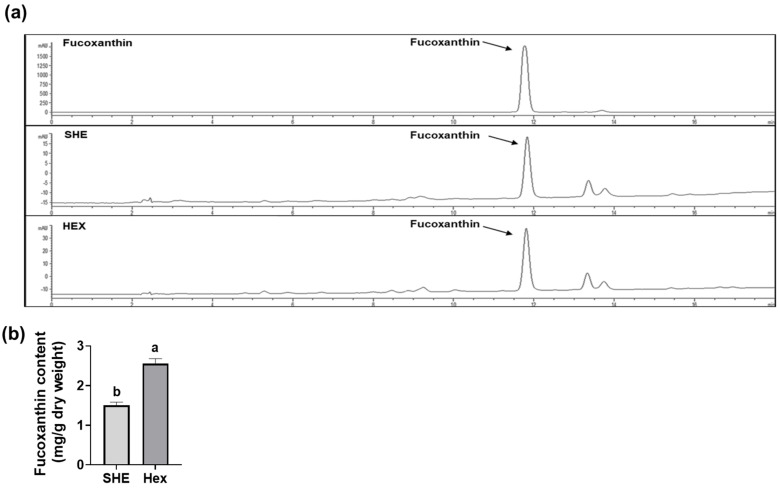
The content of fucoxanthin in SHE and its Hex fraction. (**a**) HPLC chromatograms of fucoxanthin standard, SHE, and its Hex fraction. (**b**) The content of fucoxanthin in SHE and its Hex fraction. Statistical significance was indicated by different letters within the same column (*p* < 0.05). The data are presented as means ± SD.

**Table 1 marinedrugs-21-00533-t001:** Extraction yield, total phenolic content, and total anti-oxidant capacity of the SHE and its various SHE fractions.

Extract/Fractions	Extraction Yield (%)	TPC (mg GAE/g)	ABTS (mg VCE/g)	DPPH (mg VCE/g)	FRAP (mM FSE/g)
Ethanol extract	2	7.49 ± 1.12 ^c^	15.02 ± 1.19 ^bc^	22.00 ± 3.44 ^c^	0.12 ± 0.00 ^c^
Hex fraction	35	6.72 ± 1.85 ^cd^	11.60 ± 0.76 ^cd^	21.40 ± 1.39 ^c^	0.13 ± 0.01 ^c^
CHCl_3_ fraction	3	17.49 ± 0.44 ^b^	19.75 ± 1.52 ^b^	39.58 ± 2.78 ^b^	0.15 ± 0.01 ^b^
EtOAc fraction	5	48.68 ± 3.84 ^a^	58.35 ± 5.82 ^a^	54.12 ± 3.19 ^a^	0.39 ± 0.02 ^a^
BuOH fraction	5	3.98 ± 1.04 ^cd^	7.30 ± 2.09 ^de^	13.82 ± 2.92 ^d^	0.07 ± 0.01 ^d^
H_2_O fraction	48	1.76 ± 0.15 ^d^	2.65 ± 0.76 ^e^	ND	0.04 ± 0.00 ^e^

TPC, total phenolic content; DPPH, 2,2-diphenyl-1-picrylhydrazyl radical scavenging capacity; ABTS, 2,2′-azino-bis (3-ethylbenzthiazoline-6-sulfonic acid radical scavenging capacity; FRAP, ferric-reducing antioxidant power. ND, nondetectable. The data are presented as the mean values along with standard deviation (SD) (*n* = 3). Columns that do not share a common letter are significantly different (*p* < 0.05).

**Table 2 marinedrugs-21-00533-t002:** Pearson’s correlation between TPC and three total antioxidant capacity results in SHE and its fractions.

Antioxidant Activity	TPC	ABTS Assay	DPPH Assay	FRAP Assay
TPC	1	0.95 (*p* < 0.05)	0.95 (*p* < 0.01)	0.88 (*p* < 0.01)
ABTS assay		1	0.98 (*p* < 0.01)	0.95 (*p* < 0.01)
DPPH assay			1	0.95 (*p* < 0.01)
FRAP assay				1

TPC, total phenolic content; DPPH, 2,2-diphenyl-1-picrylhydrazyl radical scavenging capacity; ABTS, 2,2′-azino-bis (3-ethylbenzthiazoline-6-sulfonic acid radical scavenging capacity; FRAP, fer-ric-reducing antioxidant power. The correlation was analyzed using Pearson’s correlation with *p* < 0.05 for a significant difference.

**Table 3 marinedrugs-21-00533-t003:** LC-MS/MS parameters for fucoxanthin.

Analyte	Precursor Ion (*m*/*z*)	Daughter Ion (*m*/*z*)	Dwell Time (s)	Cone (V)	Collision (V)
Fucoxanthin	659	109	0.091	35	30

## Data Availability

Not applicable.

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
