# Peer review of "Suppression of Pro-Inflammatory M1 Polarization of LPS-Stimulated RAW 264.7 Macrophage Cells by Fucoxanthin-Rich Sargassum hemiphyllum"

_marinedrugs, 2023, doi:10.3390/md21100533_

Round 1

Reviewer 1 Report

Interestingly, there are differences in response between RAW264.7 and BMDM cells regarding the relationship between Sargassum's antioxidant and anti-inflammatory effects, but this has not been discussed and additional discussion is required.

Minor corrections need to be made as there were some misspellings.

 1. The method in this article describes that fractions were obtained in the order of Hex, CHCl3, EtOAc, BuOH, and H2O. Please fill out Table 1 in the same order.

2. Since fucoxanthin is a fat-soluble pigment, the recovery rate is low if redissolved with water alone. It is also poorly soluble in hexane, so ethanol may have been included when distributing it using hexane. Please check the extraction and fractionation methods and make corrections.

3. The discussion describes that polarization to M1 macrophages changes Cd86 and Ccl2 mRNA expression levels in BMDMs, so these results should also be shown.

4. In M1-polarized BMDMs, Tnf mRNA expression levels significantly increased by adding SHE and Hex fractions. I feel that a discussion of this result is necessary to conclude more strongly about the anti-inflammatory effect.

5. Line 401 describes “LPS significantly increased the mRNA expression of Nfe2l2”, but in Fig. 6-b, d, the Nfe2l2 mRNA expression level in M1-polarized BMDMs is decreased compared to the control. What outcome is this argument referring to? If applicable, please indicate the figure number.

Author Response

Thank you for giving us the opportunity to submit a revised draft of our manuscript. We appreciate the time and effort that you have dedicated to providing your valuable feedback on this manuscript. We believe your suggestions have been very helpful in improving the manuscript. We have been able to incorporate changes to reflect your suggestions and they are highlighted in yellow in the revised manuscript.

Here is a point-by-point response to the reviewers’ comments and concerns.

1. The method in this article describes that fractions were obtained in the order of Hex, CHCl3, EtOAc, BuOH, and H2O. Please fill out Table 1 in the same order.

Response: Thank you for pointing this out. The order in Table 1 has been changed according to the suggestion.

2. Since fucoxanthin is a fat-soluble pigment, the recovery rate is low if redissolved with water alone. It is also poorly soluble in hexane, so ethanol may have been included when distributing it using hexane. Please check the extraction and fractionation methods and make corrections.

Response: Thanks for pointing this out. The dried samples were dissolved in methanol before fucoxanthin analysis. This part was added to the Line 517.

3. The discussion describes that polarization to M1 macrophages changes Cd86 and Ccl2 mRNA expression levels in BMDMs, so these results should also be shown.

Response: The Cd86 and Ccl22 mRNA expression levels in BMDM have been added to Figure 5, panel B.

4. In M1-polarized BMDMs, Tnf mRNA expression levels significantly increased by adding SHE and Hex fractions. I feel that a discussion of this result is necessary to conclude more strongly about the anti-inflammatory effect.

Response: Thank you for pointing this out. In the revised manuscript, the housekeeping gene has been changed from Gapdh to Rpl32, as suggested by another reviewer. Following this change, in BMDM, the Tnf mRNA expression level decreased by SHE and Hex fraction. The new data has been included in the revised manuscript Figure 5.

5. Line 401 describes “LPS significantly increased the mRNA expression of Nfe2l2”, but in Fig. 6-b, d, the Nfe2l2 mRNA expression level in M1-polarized BMDMs is decreased compared to the control. What outcome is this argument referring to? If applicable, please indicate the figure number.

Response: Sorry for the ambiguity. This sentence did refer to Figure 6. After changing the housekeeping gene, the induction of mRNA expression of Nfe2l2 was observed.

Reviewer 2 Report

In this manuscript, author studied anti-inflammatory effect of Sargassum hemiphyllum extract on RAW264.7 and bone marrow-derived macrophage cells. There is some requests for author.

 1. M1 polarization experiments

Author used RAW264.7 cells and marrow-derived macrophage cells. To induce M1 polarization, these cells were treated with LPS and INF-γ.

How did author confirm that this method was polarized in M1 macrophages?

I think it is necessary to examine the expression of specific marker proteins in M1 macrophages. Also, RAW264.7 cells are established cell line, so it is doubtful to be polarized. It is better to refer proper reference paper usd the same method, and explain the validity of this method.

 2. Page5 Figure2(b), Figure3(b)

Cybb mRNA expression was significantly suppressed in treated with SHE or Hex compared with 0 µg/mL. However, Cybb mRNA expression was lower than that of C (non treated LPS). Isn't it a problem that it is lower than the cells that are not inducing inflammation?

 3. Page6 line219; However, the expression of Tnf and Nos2 was minimal effect by SHE (Figure 4b) in M1-polarized BMDM.

Page6 line224; The induction of Il1β, Il6, and Nos2 was significantly decreased by Hex fraction in M1-polarized BMDM (Figure 5b).

It seems that the behavior of TNF-α mRNA expression in BMDM cells is different from other cytokine.  What is the reason?  It should be discussed.

 4. Page9 line 342; In the current study, SHE and its Hex fraction contained fucoxanthin as 1.50 and 2.56 mg/g extract, respectively, which responsible for ABTS and DPPH radicals scavenging activities.

Fucoxanthin content is less than 1 % of the extract. Other ingredients should be considered mainly, too. Please discuss this point.

 Minor points.

1. Page12 line471 BMDM were pretreated with SHE and its five organic solvent fractions at 0, 25, 50, and 100 μg/mL for 6 hr and then stimulated with 100 ng/mL of LPS for 24 hr.

Please describe the detail. What solvent did used to solve each fraction samples?

Ethanol or Dimethyl sulfoxide?

 2.Page 7 line252 (Figure 6a). However, Hex fraction significantly increased the expression of Sod1 and Cat in M1-polarized BMDM (Figure 6c).

The first of figure citation should be in order and the figure should put after where it first citation. Please renumber.

 3.Page9 line307; Classic M1 macrophages are primarily activated by Th1 cytokines such as LPS, IFN-γ, and TNFα.

I don't think LPS is a cytokine. Please check.

Author Response

Thank you for giving us the opportunity to submit a revised draft of our manuscript. We appreciate the time and effort that you have dedicated to providing your valuable feedback on this manuscript. We believe your suggestions have been very helpful in improving the manuscript. We have been able to incorporate changes to reflect your suggestions and they are highlighted in yellow in the revised manuscript.

Here is a point-by-point response to the reviewers’ comments and concerns.

1. M1 polarization experiments. Author used RAW264.7 cells and marrow-derived macrophage cells. To induce M1 polarization, these cells were treated with LPS and INF-γ. How did author confirm that this method was polarized in M1 macrophages? I think it is necessary to examine the expression of specific marker proteins in M1 macrophages. Also, RAW264.7 cells are established cell line, so it is doubtful to be polarized. It is better to refer proper reference paper used the same method, and explain the validity of this method.

Response: Thanks for pointing this out. The reference of M1 polarization of RAW 264.7 has been added to Line 307 and has been highlighted. “For instance, RAW 264.7 macrophages can be polarized into M1 in response to a combination of LPS and IFN-γ [23,24].” We agree with the reviewer that protein markers could be better indicator of M1 polarization. However, we did not collect protein samples in the present study. We will add this to our later research.

2. Page5 Figure2(b), Figure3(b). Cybb mRNA expression was significantly suppressed in treated with SHE or Hex compared with 0 µg/mL. However, Cybb mRNA expression was lower than that of C (non treated LPS). Isn't it a problem that it is lower than the cells that are not inducing inflammation?

Response: Thanks for pointing this out. The lower Cybb mRNA expression than that of nontreatment control has been reported in several studies. Following sentences have been added and highlight in Line 390. “SHE and Hex fraction significantly suppressed the mRNA expression of Cybb, even sur-passing the level observed in cells not exposed to LPS. NOX2 enzyme, which is encoded by Cybb gene, is involved in the M1 polarization signaling and its inhibition has been proposed to be responsible for the suppression of stress-induced oxidative burst. Sub-stances with robust antioxidant capacity have demonstrated such potent inhibitory effects in several studies [34,35].”

3. Page6 line219; However, the expression of Tnf and Nos2 was minimal effect by SHE (Figure 4b) in M1-polarized BMDM. Page6 line224; The induction of Il1β, Il6, and Nos2 was significantly decreased by Hex fraction in M1-polarized BMDM (Figure 5b). It seems that the behavior of TNF-α mRNA expression in BMDM cells is different from other cytokine.  What is the reason?  It should be discussed.

Response: Thanks for pointing this out. In the revised manuscript, the housekeeping gene has been changed from Gapdh to Rpl32, as suggested by another reviewer. Following this change, Tnf mRNA was decreased by SHE and Hex faction except for SHE at high concentration in BMDM. The following sentence has been added and highlighted in Line 324 to discuss about this. “Noticeably, Tnf mRNA level in BMDM was decreased by 25 μg/mL SHE but not 50 μg/ml SHE. This may be due to its unique constitutive decay element (CDE) [26]. At high concentration, certain substances in SHE may inhibit the degradation of Tnf mRNA in BMDM though CDE. However, further investigation is needed.”

4. Page9 line 342; In the current study, SHE and its Hex fraction contained fucoxanthin as 1.50 and 2.56 mg/g extract, respectively, which responsible for ABTS and DPPH radicals scavenging activities. Fucoxanthin content is less than 1 % of the extract. Other ingredients should be considered mainly, too. Please discuss this point.

Response: Thanks for pointing this out. Other components also exert anti-inflammatory effect. The sentence has been changed to “SHE and its Hex fraction contained fucoxanthin as 1.50 and 2.56 mg/g extract, respectively, which contribute to ABTS and DPPH radicals scavenging activities.” The following sentence has been added Line 425 and highlighted. “Other components, such as phenolic compounds and meroterpenoids, are likely to con-tribute to their anti-inflammatory effect as well.”

Minor points.

1. Page12 line471 BMDM were pretreated with SHE and its five organic solvent fractions at 0, 25, 50, and 100 μg/mL for 6 hr and then stimulated with 100 ng/mL of LPS for 24 hr. Please describe the detail. What solvent did used to solve each fraction samples? Ethanol or Dimethyl sulfoxide?

Response: Thanks for pointing this out. Sorry for the ambiguity. DMSO was used to dissolve the samples. This has been added and highlighted in Line 473.

2. Page 7 line252 (Figure 6a). However, Hex fraction significantly increased the expression of Sod1 and Cat in M1-polarized BMDM (Figure 6c). The first of figure citation should be in order and the figure should put after where it first citation. Please renumber.

Response: Thanks for pointing this out. Original figure 6a and figure 6b have been combined to make the new figure 6a. And original figure 6c and 6d have been combined to make new figure 6b.

3. Page9 line307; Classic M1 macrophages are primarily activated by Th1 cytokines such as LPS, IFN-γ, and TNFα. I don't think LPS is a cytokine. Please check.

Response: Thanks for pointing this out. Sorry for the typo. This has been corrected in the revised manuscript.

Reviewer 3 Report

In this study, Seungjin Jeong et al. investigated the antioxidant and anti-inflammatory effects of Sargassum hemiphyllum ethanol extract (SHE) and its fractions on macrophages. Among SHE's organic solvent fractions, the ethyl acetate fraction had the highest antioxidant content, while the n-hexane (Hex) fraction most effectively suppressed LPS-induced inflammation in bone marrow-derived macrophages (BMDM). Both SHE and Hex fraction notably reduced pro-inflammatory markers, particularly inhibiting M1 macrophage polarization. They also promoted antioxidant gene expression in M1-polarized BMDM. In summary, SHE and its Hex fraction modulate M1 macrophage polarization, reducing LPS-induced inflammation and oxidative stress. It has been reported that SH extract possessed antioxidant and anti-inflammatory properties. So, the novelty of this study is to study the antioxidant and anti-inflammatory effects of different components from SH.

Besides the novelty of this study, below are the major comments,

1. Many figures have Chinese text and images that are either stretched or compressed. Please standardize the format.

2. M1 and M0 macrophages have significantly different energy metabolism pathways. Using GAPDH as an internal reference for qPCR is not appropriate.

3. In WB images, the internal reference should not have multiple bands, and there is a lack of quantitative analysis.

4. I noticed that many macrophages treated with SHE, after LPS or M1 induction, showed changes in mRNA levels that were even lower than those in the control group. The author did not observe any related cell death issues. Could the author provide a detailed explanation for this?

5. If SHE can attenuate M1 polarization and this is achieved by enhancing antioxidant stress capacity, then the author should compare the effects of SHE on M2 polarization, as M1 and M2 exhibit opposing metabolic indicators.

Author Response

Thank you for giving us the opportunity to submit a revised draft of our manuscript. We appreciate the time and effort that you have dedicated to providing your valuable feedback on this manuscript. We believe your suggestions have been very helpful in improving the manuscript. We have been able to incorporate changes to reflect your suggestions and they are highlighted in yellow in the revised manuscript.

Here is a point-by-point response to the reviewers’ comments and concerns.

1. Many figures have Chinese text and images that are either stretched or compressed. Please standardize the format.

Response: We are sorry that we could not see these Chinese text and images on our computers. If possible, could you take a snapshot of the images and send us?

2. M1 and M0 macrophages have significantly different energy metabolism pathways. Using GAPDH as an internal reference for qPCR is not appropriate.

Response: Thank you for pointing this out. In the revised manuscript, we have used Rpl32 instead of Gapdh in the mRNA analysis. The figures and content have been changed correspondingly. We have noticed a significant change of the results, especially the antioxidant gene expressions, after switching the housekeeping gene. We deeply appreciate your insightful suggestions.  

3. In WB images, the internal reference should not have multiple bands, and there is a lack of quantitative analysis.

Response: Thanks for pointing this out. The internal control (ACTIN) have been analyzed again and the new images have been used in the revised manuscript. And quantitative analysis has been added.

4. I noticed that many macrophages treated with SHE, after LPS or M1 induction, showed changes in mRNA levels that were even lower than those in the control group. The author did not observe any related cell death issues. Could the author provide a detailed explanation for this?

Response: Thanks for pointing this out. We have not observed any noticeable cell death at the indicated concentrations. After switching the housekeeping gene, the reduction of mRNA levels lower than in the control group in M1 has not been observed. We believe, as you suggested, Gapdh was not an appropriate housekeeping gene in M1, which led to the abnormally low level of mRNA in M1 macrophages. Regarding LPS induction, Cybb was the one lower than non-LPS samples. The following sentences have been added and highlight in Line 390. “SHE and Hex fraction significantly suppressed the mRNA expression of Cybb, even sur-passing the level observed in cells not exposed to LPS. NOX2 enzyme, which is encoded by Cybb gene, is involved in the M1 polarization signaling and its inhibition has been proposed to be responsible for the suppression of stress-induced oxidative burst. Sub-stances with robust antioxidant capacity have demonstrated such potent inhibitory effects in several studies [34,35].”

5. If SHE can attenuate M1 polarization and this is achieved by enhancing antioxidant stress capacity, then the author should compare the effects of SHE on M2 polarization, as M1 and M2 exhibit opposing metabolic indicators.

Response: Thanks for pointing this out. We agree that the regulation of SHE on M2 polarization is a critical component in studying the immunoregulatory impact of SHE, however it cannot be analyzed in this manuscript due to current resource constraints. Nevertheless, we intend to investigate the influence of SHE on M2 polarization in our future research endeavors.

Round 2

Reviewer 1 Report

I believe that the reviewer's comments have been appropriately addressed and there are no additional major revisions.

Minor points
Add “(Figure 6b)” to line 224.

Reviewer 2 Report

I checked revised manuscript.

Author adequately responded to reviewer comments.

Reviewer 3 Report

No more comments